# Characterization of Host-Specific Genes from Pine- and Grass-Associated Species of the *Fusarium fujikuroi* Species Complex

**DOI:** 10.3390/pathogens11080858

**Published:** 2022-07-29

**Authors:** Claudette Dewing, Magrieta A. Van der Nest, Quentin C. Santana, Robert H. Proctor, Brenda D. Wingfield, Emma T. Steenkamp, Lieschen De Vos

**Affiliations:** 1Department of Biochemistry, Genetics and Microbiology, Forestry and Agricultural Biotechnology Institute (FABI), University of Pretoria, Private Bag X20, Pretoria 0028, South Africa; claudette.dewing@up.ac.za (C.D.); vandernestm@arc.agric.za (M.A.V.d.N.); quentin.santana@up.ac.za (Q.C.S.); brenda.wingfield@up.ac.za (B.D.W.); emma.steenkamp@up.ac.za (E.T.S.); 2Biotechnology Platform, Agricultural Research Council, Onderstepoort, Pretoria 0110, South Africa; 3Mycotoxin Prevention and Applied Microbiology Unit, United States Department of Agriculture (USDA), Agriculture Research Service, National Center for Agricultural Utilization Research, 1815 N. University St., Peoria, IL 61604, USA; robert.proctor@usda.gov

**Keywords:** *Fusarium fujikuroi* species complex, *Fusarium circinatum*, *Fusarium temperatum*, pitch canker, comparative genomics, host-specificity, horizontal gene transfer, subtelomeres

## Abstract

The *Fusarium fujikuroi* species complex (FFSC) includes socioeconomically important pathogens that cause disease for numerous crops and synthesize a variety of secondary metabolites that can contaminate feedstocks and food. Here, we used comparative genomics to elucidate processes underlying the ability of pine-associated and grass-associated FFSC species to colonize tissues of their respective plant hosts. We characterized the identity, possible functions, evolutionary origins, and chromosomal positions of the host-range-associated genes encoded by the two groups of fungi. The 72 and 47 genes identified as unique to the respective genome groups were potentially involved in diverse processes, ranging from transcription, regulation, and substrate transport through to virulence/pathogenicity. Most genes arose early during the evolution of *Fusarium*/FFSC and were only subsequently retained in some lineages, while some had origins outside *Fusarium*. Although differences in the densities of these genes were especially noticeable on the conditionally dispensable chromosome of *F. temperatum* (representing the grass-associates) and *F. circinatum* (representing the pine-associates), the host-range-associated genes tended to be located towards the subtelomeric regions of chromosomes. Taken together, these results demonstrate that multiple mechanisms drive the emergence of genes in the grass- and pine-associated FFSC taxa examined. It also highlighted the diversity of the molecular processes potentially underlying niche-specificity in these and other *Fusarium* species.

## 1. Introduction

The genus *Fusarium* represents a diverse group of Ascomycetes that are important in industry, agriculture, and medicine [1,2]. While known species in the genus include pathogens of humans [3,4,5], animals [6] and insects [7,8], the majority are destructive plant pathogens [9,10,11,12,13,14]. Additionally, *Fusarium* species generate a variety of toxic secondary metabolites that can contaminate feedstocks and food, affecting the quality and quantity of most agriculturally important crops [15]. Due to their importance in forestry, agriculture and medicine, the genomes of many *Fusarium* species have been determined [16].

*Fusarium* species associate with diverse plant species, ranging from gymnosperms to angiosperms, including many monocots in the grass family, Poaceae [17,18,19]. Examples of these species are included in the phylogenetically defined lineage of *Fusarium*, known as the *Fusarium fujikuroi* species complex (FFSC) [1,2]. FFSC taxa that associate with gymnosperms (e.g., those in the pine family Pinaceae) include *Fusarium circinatum*, an economically important pathogen of *Pinus* species, as well as *Fusarium fracticaudum* and *Fusarium pininemorale* that colonize the tissue of these plants without any apparent disease symptoms [11,13]. Examples of grass-host-associated species include the prairie grass endophyte *Fusarium konzum* [20] and the maize pathogens *Fusarium subglutinans* and *Fusarium temperatum* [10,21]. However, these associations are not always strict, as in the case of *F. circinatum*, which can also colonize maize and grasses (from Poaceae) [22,23,24,25], various herbaceous dicots (besides Poaceae) [26], as well as certain other non-pine conifers [27,28]. Despite their diverse host associations, these six species are closely related [13]. Their genomes are characterized by high levels of synteny [29,30], and some species can hybridize [30,31]. The ability of *F. circinatum* and *F. temperatum* to hybridize suggests that since they diverged from a recent common ancestor, they have co-evolved with their respective plant hosts (*Pinus* spp. and maize) in overlapping geographic ranges [10,18].

Advances in genomics have enhanced the discovery and analysis of the mechanisms of host-specificity and processes underlying plant–fungus interactions [32]. These data showed that fungal genomes, including those of *Fusarium*, are divided into core and accessory compartments [33,34]. Core genes typically play an integral part in primary metabolism, reproductive strategies, and development. Accessory genes, on the other hand, are typically non-essential for growth and development but encode for adaptive and niche-associated traits, such as pathogenicity and virulence factors [34]. Among closely related *Fusarium* species, the core compartment is usually conserved, syntenous and gene rich [14], whereas the accessory compartment is gene-poor and often also includes subtelomeric regions and supernumerary/accessory chromosomes [35]. Both the latter are typically associated with loss of synteny and have been linked to genes promoting the adaptation and survival of pathogens [36].

Previous comparative genomics studies have enhanced our knowledge of how genome evolution affects host-specificity and pathogenicity [37,38,39]. Various evolutionary processes are now known to shape the outcome of the plant–fungus interaction [40,41,42,43]. Additionally, genes or genomic regions that are not under selection can be lost or accumulate mutations, which can influence the adaptation of fungi to different host environments [44]. However, little information regarding such processes is available for the FFSC, especially those determining their ability to colonize certain plant tissues.

In the current study, we exploited the publicly available whole genome sequences for FFSC species (NCBI; https://www.ncbi.nlm.nih.gov/ (accessed on 1 December 2021)) to study the molecular processes that may underly the association of these fungi with their plant hosts. For this purpose, we employed a comparative genomics approach using the data available for six closely related FFSC species associated with pine (i.e., *F. circinatum, F. fracticaudum* and *F. pininemorale*) and grass (i.e., *F. konzum, F. subglutinans* and *F. temperatum*). Our specific aims were to identify the genes that are unique to each group of species, and then characterize the genes in terms of ontology, predicted pathways and mechanisms, chromosomal location, and potential ancestral origin. Our findings will contribute to the current understanding of the diversity and evolution of these species.

## 2. Results

### 2.1. Genome Sequences, Assembly Completeness and Gene Prediction

The three genomes for pine-host-associated species were from *F. circinatum* FSP34, *F. fracticaudum* CMW 25245 and *F. pininemorale* CMW 25243. Genomes for grass-host-associated species were from *F. konzum* NRRL 11616, *F. subglutinans* NRRL 66333 and *F. temperatum* CMW 40964.

Analysis with the Benchmarking Universal Single-Copy Orthologs (BUSCO) tool v. 3.0.2 [45] using the “Sordariomyceta” database, showed that all six genomes were from 97.3% to 99.1% complete (Appendix A). These genomes were also similar in size, gene density, and G + C content (Table 1). Comparison of the four genomes assembled into pseudochromosomes (*F. circinatum, F. fracticaudum, F. pininemorale* and *F. temperatum*) further showed that chromosome size seemed to be conserved throughout Chromosomes 1–11, as they did not differ by more than a factor of 1.00–1.10 (Appendix A). However, Chromosome 12 showed extreme chromosome length polymorphism (CLP) and differed by as much as a factor of 2.08, ranging from 0.5 Mb in *F. circinatum* to 1.1 Mb in *F. fracticaudum* (Appendix A).

For further analysis, we used the *F. circinatum* genome as a reference for pine-host-associated species and the *F. temperatum* genome as a reference for grass-host-associated species. Analysis of the telomere-associated repeat sequence (TTAGGG/CCCTAA) [47,48] in the reference genomes indicated that most of the chromosomes had a telomeric cap (Appendix A). In the *F. circinatum* assembly, seven chromosomes had two telomeric caps and five had only one telomeric cap. In the *F. temperatum* assembly, nine chromosomes had two telomeric caps, two chromosomes had one telomeric cap and one chromosome had no telomeric cap.

### 2.2. Genes Unique to Pine- and Grass-Host-Associated Species

Analysis with OrthoFinder v. 2.3.1 [49] facilitated the identification of 72 and 47 genes that were unique to pine- or grass-host-associated *Fusarium* species, respectively (Appendix A). Multiple paralogs of a few of these genes occurred in some genomes: one paralog of one gene in *F. circinatum* and *F. pininemorale*, two paralogs of two genes in *F. fracticaudum* and *F. subglutinans* and three paralogs of one gene in *F. konzum*. Interestingly, the paralogs of a particular gene were present together on the same chromosome, except in *F. fracticaudum*, where two of these genes were located on different chromosomes. No inferences could be made regarding the chromosomal location of the multiple-copy genes of *F. konzum* and *F. subglutinans* due to the fragmented nature of these assemblies. The phylogenetic relationship between the paralogous genes and the unique gene could be determined for both *F. fracticaudum* paralogs, the one *F. pininemorale* paralog and only one of the two *F. subglutinans* paralogs. The phylogenetic relationship for the remaining *F. subglutinans* paralog, along with the three *F. konzum* paralogs, could not be determined, as the ancestral origins of these genes were unknown and/or had no significant BLAST hit in the NCBI database (Appendix A). All paralogs shared the same ancestral origins as the unique gene, except for *F. fracticaudum.* The paralog of the unique gene was located on a different chromosome and grouped within the ancestral origin group FFSC and FOSC, whereas the unique gene shared an ancestral origin outside *Fusarium* but in the Ascomycetes.

Most of the host-range-associated genes had a gene ontology (GO) description available from Blast2GO [50]. Of the 72 unique *F. circinatum* genes, no GO terms were available for 11 of the genes. Apart from these 11 genes, 26 genes were either hypothetical or predicted to encode uncharacterized proteins. The remaining 35 could be organized into four broad groups based on the proteins they encode, i.e., those involved in virulence, transcriptional regulators, substrate transporters and permeases, proteins potentially involved the metabolism of carbohydrates, fatty acids, and steroids (Appendix A). Of the 47 unique *F. temperatum* genes, five did not have a hit in BLAST analysis. A further 17 unique *F. temperatum* genes were either hypothetical or predicted to encode uncharacterized proteins. The remaining 25 genes could be organized into five broad groups based on the likely functions of their protein products, i.e., those involved in virulence, substrate transporters and permeases, proteins potentially involved the metabolism of carbohydrates, fatty acids, amino acids, and steroids, and an HET-domain protein (Appendix A).

To determine whether certain GO terms in the respective unique sets were significantly enriched relative to the rest of each respective genome, we employed Fisher’s Exact test implemented in Blast2GO [50]. In the pine-host-associated set, five GO terms were significantly (*p* < 0.05) enriched in comparison to the whole genome (Appendix A), whereas 21 GO terms were significantly (*p* < 0.05) enriched in the grass-host-associated set (Appendix A). Of the genes associated with these enriched GO terms, one gene was classified as having a “biologically relevant” role (i.e., glutamate metabolism), and four as having “essential molecular functions”, such as coding for a gene product involved in glutamate decarboxylase activity, steroid dehydrogenase activity and RNA-DNA hybrid ribonuclease activity. Furthermore, one gene was coded for a gene product involved in cellular components, such as the microtubule and kinesin complex.

We also used previous expression data for *F. circinatum* [51,52] to search for evidence of expression of the pine-host-associated genes (Appendix A). The combined results indicated that 62 of the 72 pine-host-associated genes were transcribed under the conditions used to generate the expression data [51,52,53]. No expression data are currently available for any of the grass-host-associated *Fusarium* species. 

### 2.3. Phylogenetic Origins of the Host-Range-Associated Genes

Ancestry of genes in the pine- and grass-host-associated sets were inferred using their protein sequences in BLAST searches against NCBI’s non-redundant protein database analyses (performed on 04 October 2021), which were supplemented in some cases by several rounds of alignment of the retrieved sequences followed by phylogenetic analyses. Based on these data, we assigned the 119 genes from the two sets into eight groups (Table 2 and Appendix A). Genes included in the first seven groups had evolutionary origins within the FFSC, the broader *Fusarium* genus, or lineages outside *Fusarium* but within the Ascomycetes. The last group contained those genes lacking significant BLAST hits in the NCBI database, as well as those that did not comply with the cut-off values stated in Section 4.3.

The results of these analyses indicated that the ancestral origin for most of the genes was in members of the FFSC or other lineages of *Fusarium*. Indeed, based on their phylogenies, the genes included in groups 1–4 (i.e., 39 genes in the pine-associated set and 25 in the grass-associated set) likely emerged early during the evolution of *Fusarium* and/or the FFSC, after which they were retained in only some lineages. By contrast, 12 pine-host-associated and seven grass-host-associated genes lacked homologs in *Fusarium* and were more closely related to homologs in other Ascomycetes (groups 5 and 6) or outside of the kingdom Fungi (group 7). A considerable proportion of the genes had no detectable homologs in GenBank.

### 2.4. Genomic Distribution of the Host-Range-Associated Genes

All *F. circinatum* chromosomes included at least one pine-host-associated gene, while all but two *F. temperatum* chromosomes included at least one grass-host-associated genes (Figure 1, Figure 2, Appendix A). Chi-squared tests indicated that significantly (*p* < 0.05) more of these genes were located in subtelomeric regions than outside of them (Table 2). That is, about half of the host-associated genes in each species were restricted to subtelomeric regions, which constitute approximately one quarter of each genome (Appendix A). Of the 38 host-range-associated genes in subtelomeric regions of *F. circinatum*, 25 yielded significant hits in BLAST analysis (and evidence of expression; see Appendix A). By contrast, 16 of the 30 host-range-associated genes in subtelomeric regions in *F. temperatum* had BLAST hits. Phylogenetic analyses of these genes indicated that most had evolutionary origins within fungi (see Table 2). However, the origins of a substantial number of subtelomeric genes remained unknown, as there were no homologs for them in GenBank, while one gene in both sets apparently has origins outside of fungi (see groups 7 and 8, respectively, in Table 2).

The density of host-range-associated genes differed among chromosomes in both *F. circinatum* and *F. temperatum* (Appendix A). Most chromosomes, except for Chromosome 5, 10 and 12, had higher unique gene densities for *F. temperatum* when the assemblies for *F. circinatum* and *F. temperatum* were compared. Homologous chromosomes of *F. circinatum* and *F. temperatum* had similar densities of host-range-associated genes, ranging from 1.04× (Chromosome 4) to 1.97× (Chromosome 7). Moderate differences appeared in the host-range-associated gene density (more than two-fold) of Chromosome 9 and 12, where *F. circinatum* contained more than *F. temperatum*, and vice versa, for these two chromosomes. Lastly, a major difference (more than seven-fold) occurred on Chromosome 3 and 6, where *F. circinatum* had more host-range-associated genes compared to the same chromosomes in *F. temperatum*. The sequence of *F. circinatum* Chromosome 6 did not include a telomeric cap. It is also worth noting that Chromosome 12 in *F. temperatum* contained more host-range-associated genes compared to the same chromosome in *F. circinatum*.

Our data showed that some of the host-range-associated genes occurred in clusters or were located in close proximity to one another, and that such genes could have the same or different ancestral origins (Figure 3; Appendix A). For example, *F. circinatum* genes FCIR_6_gene_2.26 and FCIR_6_gene_2.131 were adjacent to one another on Chromosome 6 and were in the same ancestral origin group (Group 5) (Figure 3A; Table 2), whereas genes FCIR_6_gene_2.2 and FCIR_6_gene_2.142 were near one another on Chromosome 6 but were in different ancestral origin groups (Groups 8 and 7, respectively) (Figure 3A,B; Table 2).

Using SynChro [54], we investigated whether gene areas flanking clustered host-range-associated genes exhibited synteny in *F. circinatum* and *F. temperatum* (Appendix A). For this analysis, we included 10 genes flanking host-range-associated genes, five genes upstream and five downstream. We used the analysis to determine the frequency with which 1–10 genes flanking host-range-associated genes were syntenic in *F. circinatum* and *F. temperatum* These analyses indicated that the genomic locations around host-range-associated genes in *F. circinatum* were normally distributed (Shapiro–Wilk’s test for departure from normality; *p* > 0.05; W = 0.95) [55], but not in *F. temperatum* (*p* < 0.05; W = 0.86) (Figure 4).

The difference in distribution pattern was particularly evident when comparing clusters in the two genomes that had five and six of the 10 syntenic genes. In *F. temperatum*, the clusters with five out of 10 syntenous genes appeared to be severely depleted, indicating minimal synteny when compared to the *F. circinatum* genome. However, the clusters with six out of 10 syntenous genes in *F. temperatum* were expanded compared to *F. circinatum*. The general trend was that genes flanking host-range-associated genes in *F. circinatum* were frequently syntenic with genes in *F. temperatum*, whereas genes flanking host-range-associated genes in *F. temperatum* were less frequently syntenic with genes in *F. circinatum.* This trend likely suggests higher conservation of the genes and/or genomic regions neigbouring or flanking the position of host-range-associated genes in *F. circinatum* compared to *F. temperatum.*

## 3. Discussion

In the current study, we used a comparative genomics approach to explore the molecular basis of plant–fungus interactions by making use of two groups of *Fusarium* species in the FFSC, i.e., one associated with pine and the other with grass. This allowed for the identification of sets of genes that are unique to each of the two groups of *Fusarium* species, and that markedly differed in terms of their identity and the function of the proteins they encode. The two sets of genes also showed large differences in their ancestral origins, and they tended to occur in subtelomeric regions of chromosomes. Overall, these genes highlighted the different evolutionary origins and molecular processes that underpin the capacity of these fungi to colonize their respective plant hosts.

To conduct this study, we used two high-quality reference genome assemblies. The one was for the pine pathogen *F. circinatum*, while the other was for the maize pathogen *F. temperatum*. Within these assemblies, most chromosomes had telomere-to-telomere coverage, which allowed us to compare the localization and clustering of the identified host-range-associated genes. This also allowed for comparisons of chromosomal architecture among four of the six studied *Fusarium* species.

The host-range-associated genes identified in the genomes of the pine-host-associates (*F. circinatum, F. fracticaudum* and *F. pininemorale*) and in those of the grass-host-associates (*F. konzum, F. subglutinans* and *F. temperatum*) were preferentially localized in subtelomeric regions, possibly due to the greater genomic instability occurring in these regions compared to elsewhere on the chromosome [56]. It has been speculated that the preferred subtelomeric location facilitates gene-switching and expression, and the generation of new gene variants [57,58]. The subtelomeric location of genes has been studied in several organisms, which include *Homo sapiens* [59], *Drosophila melanogaster* [60], *Plasmodium falciparum* [61], *Trypanosoma brucei* [62], *Saccharomyces cerevisiae* [63], and the fungal parasite *Encephalitozoon cuniculi* [64]. Upon genome analyses of *S. cerevisiae*, Louis [63] uncovered that subtelomeric genes belong to several gene families encoding proteins involved in carbon source utilization. In humans, for example, genes encoding for olfactory receptors (i.e., proteins capable of binding odor molecules) are the most studied and largest gene family located in the subtelomeric regions of the human genome [65,66].

The presence of subtelomeric regions provide an evolutionary advantage and have potential importance in replication and chromosome stability, but also seem to play important roles in the reversible silencing of genes mediated by proteins binding to the telomere, and engagement in ectopic recombination with other subtelomeres, likely resulting in gene diversification [58,67,68]. The subtelomeric regions are also thought to have higher genomic instability than elsewhere on the chromosome, which possibly allows for the tight regulation, and occasional switching of expression between different gene copies, allowing for organisms to evade the immune system of their respective hosts [67,68]. In addition, the literature suggests that genes located in subtelomeres play important evolutionary roles towards adaptation, such as contributing to niche- or host-specificity and virulence in pathogens [69,70,71,72]. For example, the fungus *Pneumocystis jiroveci* has large and highly variable gene families located in its subtelomeric regions that encode for surface proteins with tightly regulated, yet switchable, expression patterns [73]. *Fusarium* subtelomeres have been reported to contain virulence genes involved in host-specificity, e.g., in *F. graminearum* [36,74,75] and *F. fujikuroi* [29]. In *F. circinatum*, a locus determining growth-rate region was also characterized from a subtelomeric region [71]. In the case of the *Fusarium* species examined in this study, a large proportion of the genes inferring potential host-specificity were positioned on subtelomeres, pointing to their possible roles in fungal development, adaptation, and survival for these *Fusarium* species.

The presence/absence of host-range-associated genes in the two sets of genomes, as well as their placement in subtelomeric regions, is consistent with what is expected for genes belonging to the accessory genomic compartment, since these regions are considered to be components of the accessory genome. In other words, they represent accessory genes likely involved in niche-associated traits, which are typically enriched for functions related to cell–cell interactions, secondary metabolism, and stress responses, all potential contributors to pathogenicity and virulence [34,76]. The dynamic evolutionary processes associated with the accessory genome [34,41,42,77,78] likely allowed for the preferential loss and/or gain of genes with new functions. This ultimately led to the pine- and grass-host-associates examined here being able to colonize and survive on the tissue of their respective hosts.

Our findings are consistent with the notion that horizontal gene transfer (HGT) plays a major role in the evolution of accessory genomes [34,41,42,77,78,79,80,81]. As such, HGT is increasingly recognized as a significant driver of adaptive evolution in fungi and the pathogenic traits of pathogens [82,83,84]. For example, the ability of *Valsa mali* to grow on its host and be pathogenic to it was acquired via HGT from diverse sources [84]. HGT can even cause basic lifestyle changes, e.g., facilitating the transition of the plant-associated ancestor of *Metarhizium* into the entomopathogenic taxon known today [85]. Among FFSC species, multiple independent HGT events were reported to cause the development of the growth-determining locus in *F. circinatum* [71]. The polyphyletic origins of many of the host-range-associated genes identified in the current study were likely also acquired from such diverse external sources (see groups 5–8 in Table 2), while most of the remainder likely originated within *Fusarium* or the FFSC and were retained only in certain lineages (groups 1–4 in Table 2). More research is needed to determine the possible role of internal genomic mutations (due to duplication, displacement, and translocation events) [30,86,87] in determining the host associations of the examined fungi.

Despite the existence of genes unique to the respective pine- and grass-host-associated fungi, their genomes were remarkably syntenic [29,30]. Variability was most evident when comparisons of inversions were investigated. Inversions closer to the ends of chromosomes were previously reported in fungi [36,88,89]. Inversions were most noticeable for Chromosome 8, especially when considering the host-range-associated genes on this chromosome. All host-range-associated genes from Chromosome 8 were located in regions prone to inversion. The variability of chromosome 8 is likely due to the sizable reciprocal translocation with Chromosome 11 within some FFSC taxa, including all species investigated in this study [30,90,91,92,93]. Chromosome 11 from *F. circinatum* had only one unique gene located outside of the inversed regions, whereas the same chromosome in *F. temperatum* lacked any host-range-associated genes. This suggests that Chromosome 8 is more predisposed to chromosomal structural changes. Current advances in sequencing technologies highlighted the importance of chromosome rearrangements events in cellular processes by providing longer reads and telomere-to-telomere coverage [94]. Translocation events have been reported to drive the evolution of secondary metabolite biosynthetic gene clusters, whereas chromosome rearrangements may result in gene loss and/or gain, genetic diversity, and changes in pathogenicity [95,96,97,98]. For example, effector genes located in rearranged chromosomal regions are more prone to evolution and are more likely to contribute to adaptation and speciation [99,100].

The importance of Chromosome 12 in possessing host-range-associated genes poses an interesting question. This chromosome is a dispensable chromosome within the FFSC [42,101,102,103] and can be strain-specific [29,102]. Chromosome 12 is not essential for pathogenicity [29,102,104] and its role in niche-specificity is unknown. However, the larger Chromosome 12 of *F. temperatum* contained more host-range-associated genes that were potentially involved in niche-specificity among *Fusarium* species that are associated with Poaceae. This occurrence may imply that some elements of the chromosome length polymorphism between *F. circinatum* and *F. temperatum* accounts for the addition of host-range-associated genes and that this chromosome is rapidly evolving. Chromosome 12 requires further investigation and would, therefore, be an important target for future studies.

Based on their inferred products, the host-range-associated genes could mostly be grouped as being involved in virulence, substrate transport, and metabolism of carbohydrates, and in the case of *F. temperatum*, also as a heterokaryon incompatibility (HET)-domain protein. It also highlights the evolutionary changes that Poaceae-associated *Fusarium* species had to undergo for these genes to be successful colonizers of members of Poaceae. The presence of these groups of genes is indicative of their role in the survival and adaptation of these *Fusarium* species, likely providing a competitive advantage as well as a protective advantage against the host defense responses.

In conclusion, this study used near-complete and high-quality genome data to identify a set of genes that potentially underpin the association of six *Fusarium* species with their pine and grass hosts. These genes are involved in diverse processes that could potentially provide the fungi with ecological advantages in their respective niches [105,106,107,108,109,110]. Furthermore, these genes preferentially form part of the accessory genome of the examined fungi, where diverse processes likely determined their evolution and development. This work thus forms a strong foundation from which future studies could explore the functional relevance of the identified genes, as well as their expression, especially given their chromosomal locations. Such studies are needed if we are to understand the ecology and biology of the fungi that threaten the health of socioeconomically important plants and crops.

## 4. Materials and Methods

### 4.1. Genome Sequences, Assembly Completeness and Gene Prediction

The completeness of each genome assembly was evaluated through the Benchmarking Universal Single-Copy Orthologs (BUSCO) tool v. 3.0.2 [45] using the “Sordariomyceta” database. Using the significant synteny found within the FFSC [29,30], scaffolds from the genomes of *F. pininemorale* (isolated from diseased *P. tecunumanii* tissue [13,91]) and *F. fracticaudum* (isolated from diseased *Pinus maximinoi* tissue [13,92]) were compared against the genomes of *F. circinatum* [93] (isolated from diseased *Pinus radiata* tissue [11,111]), *F. fujikuroi* [29] and *F. temperatum* [90] (isolated from diseased teosinte tissue [10,90]), as these have been assembled into chromosomes. The scaffolds were ordered and orientated into 12 contiguous pseudochromosomes (representing the core chromosomes (1–11) and the accessory chromosome (12)) [30]. The genomes of *F. subglutinans* (733 contigs; isolated from diseased maize tissue [21,112]) and *F. konzum* (2262 contigs; isolated from prairie grass [20,112]) were considered too fragmented to be assembled into pseudochromosomes.

A reference genome was selected to represent either the pine- or grass-host-associates. The purpose of this was to achieve a detailed comparison with two well-assembled genomes between *Fusarium* species colonizing different plant hosts. *Fusarium circinatum* was chosen to represent the pine-host-associates and *F. temperatum* represented grass-host-associates. The abundance of the telomere-associated repeat sequence (TTAGGG/CCCTAA) [47,48] was evaluated using a motif search performed in CLC Genomics Workbench v. 11 (CLC bio, Aarhus, Denmark) to further investigate the completeness of the two reference genomes (*F. circinatum* and *F. temperatum*). For the motif search, a window size of 10,000 bp with 5000 bp increments was used. Only repeats with ≥80% similarity to the telomere repeat were considered in this analysis.

The MAKER2 pipeline [113] was utilized for structural annotations of all six genomes in order to identify protein-coding genes. Gene prediction was performed in MAKER using SNAP [114], GeneMark ES [115] and AUGUSTUS [116]. As additional evidence, gene model data from *F. circinatum* [93], *F. fujikuroi* [29], *F. verticillioides* and *F. graminearum* [42], as well as *F. mangiferae* and *F. proliferatum* [19], were included. These isolates were selected due to the availability of their genomic information on the NCBI public database (https://www.ncbi.nlm.nih.gov/; (accessed on 1 December 2021).

### 4.2. Genes Unique to Pine- and Grass-Host-Associated Species

The gene content of all six *Fusarium* genomes was evaluated to determine which genes are unique to the species associated with the two groups of plant hosts. For this purpose, OrthoFinder v. 2.3.1 (https://github.com/davidemms/OrthoFinder (accessed on 1 December 2021)) was implemented [49]. The genes occurring only in the genomes of the pine- and grass-host-associates were labeled as “unique” genes.

Functional annotation was performed using the Blast2GO v. 6 [50] plugin (Valencia, Spain) for CLC Genomics Workbench. A two-tailed Fisher exact test was implemented (*p* < 0.05) in Blast2GO [50] to detect the GO terms that were overrepresented in the host-range-associated genes set of both *Fusarium* representatives, using the whole genome of each as reference. 

### 4.3. Phylogenetic Origins of the Host-Range-Associated Genes

The two unique gene sets were uploaded onto NCBI to perform BLASTp searches against the non-redundant database using the online position-specific iterative (PSI) BLAST tool (https://blast.ncbi.nlm.nih.gov/Blast.cgi (accessed on 1 December 2021)) [117]. All highly divergent protein sequences were excluded by considering only those sequences with at least 40% amino acid identity over 70% of the query sequence length and that had E-values ≤ 1 × 10^−5^ [71]. All the sequences were aligned using the constraint-based alignment tool (COBALT) [118] and phylogenetic trees were viewed to determine the ancestral origins of the host-range-associated genes. The host-range-associated genes with unclear ancestral origin were selected to construct phylogenetic trees. To infer phylogenies, all sequences were aligned with Multiple sequence Alignment based on Fast Fourier Transform (MAFFT) v. 7.0 with default settings [119]. These alignments included the relevant *Fusarium* sequences, together with those from other Ascomycota. MEGA v. 7.0.26 (https://www.megasoftware.net/ (accessed on 1 December 2021)) [120] was used to draw initial tree(s) for the heuristic search by applying Neighbor-Join and BioNJ algorithms to a matrix of pairwise distances to estimate the best substitution model to use. The Maximum Likelihood branch support was estimated using bootstrap analyses based on 100 pseudoreplicates. 

### 4.4. Genomic Distribution of the Host-Range-Associated Genes

The location and distribution of the host-range-associated genes were plotted across the 12 chromosomes in each of the two reference genomes using KaryoploteR v. 3.9 (http://bioconductor.org/packages/karyoploteR (accessed on 1 December 2021)) [121]. The difference in the distribution of genes in the different genomic regions was evaluated using Chi-squared tests (*p* < 0.05), by comparing those within and outside subtelomeric regions, with the null hypothesis that they are characterized by similar frequencies. Here, we regarded the first and last 500 kb from chromosome ends as subtelomeres [71,122,123,124]. Furthermore, the synteny and conservation of the location of the different host-range-associated genes were studied with SynChro which revealed the synteny breakpoints between the reference and query genomes [54]. Lastly, the Shapiro–Wilks test was performed to test for normality by detecting all departures from normality for genes belonging to *F. circinatum* and *F. temperatum*.

## Figures and Tables

**Figure 1 pathogens-11-00858-f001:**
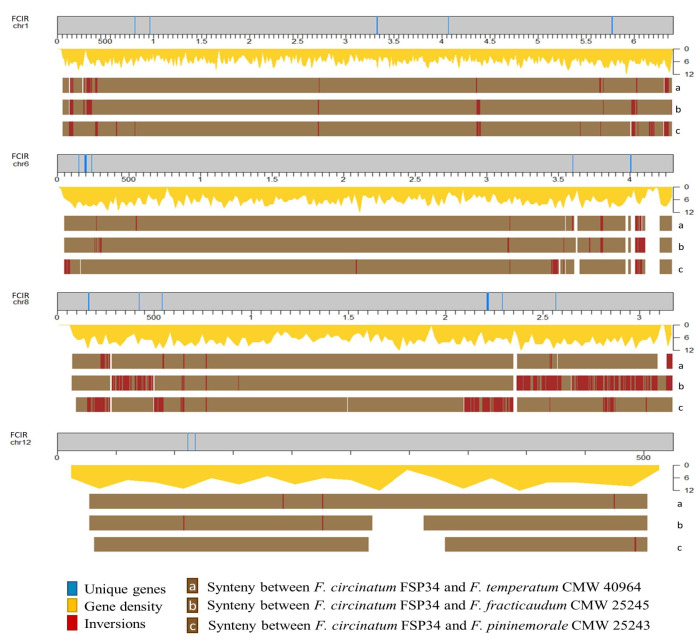
The distribution of host-range-associated genes from pine-host-associated *Fusarium* species and conservation of synteny across and among chromosomes and genomes. Distribution of pine-associated genes across each of the chromosomes are indicated by vertical blue lines. The conservation of synteny and inversion between the relevant genomes are indicated in the brown blocks and red lines. FCIR = *F. circinatum*; chromosome size is given in Kb.

**Figure 2 pathogens-11-00858-f002:**
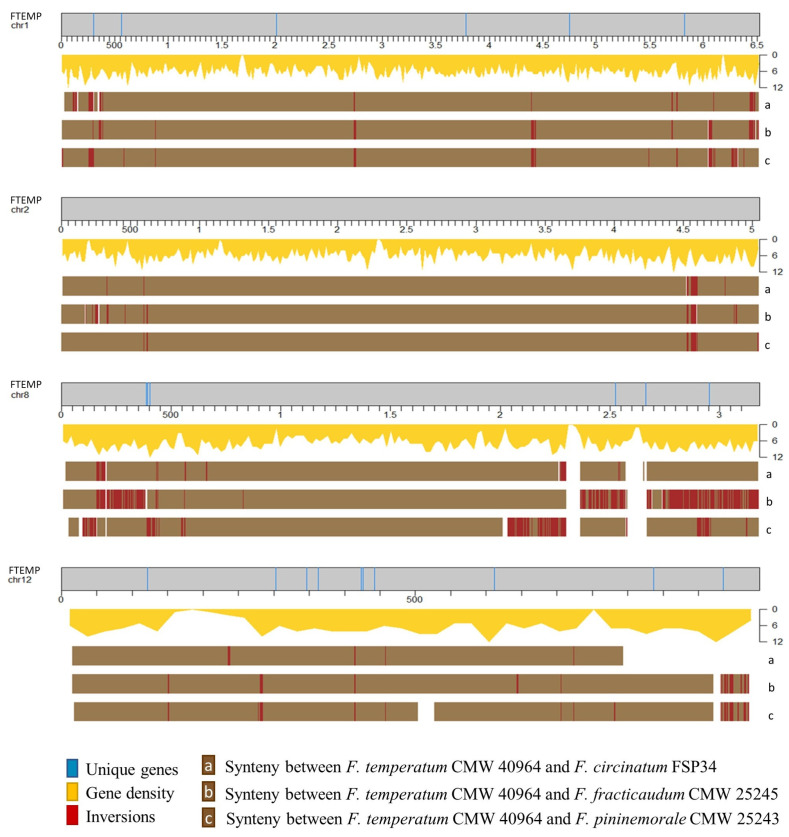
The distribution of host-range-associated genes from Poaceae-host-associated *Fusarium* species and conservation of synteny across and between chromosomes and genomes. Poaceae-associated genes distribution across each of the chromosomes as indicated by the blue lines. The conservation of synteny and inversion between the relevant genomes is indicated in the brown blocks and red lines. FTEMP = *F. temperatum*; chromosome size is given in Kb.

**Figure 3 pathogens-11-00858-f003:**
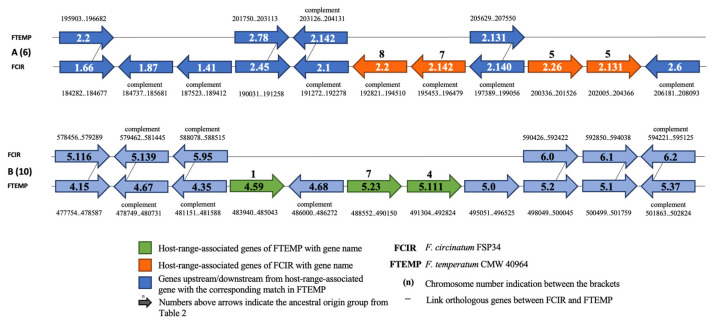
Examples of clusters of host-range-associated genes on Chromosome 6 of *F. circinatum* (**A**) and Chromosome 10 of *F. temperatum* (**B**). Numbers within the block arrows indicate gene identities. Bold letters above the green and orange genes indicate ancestral origin group as described in Table 2.

**Figure 4 pathogens-11-00858-f004:**
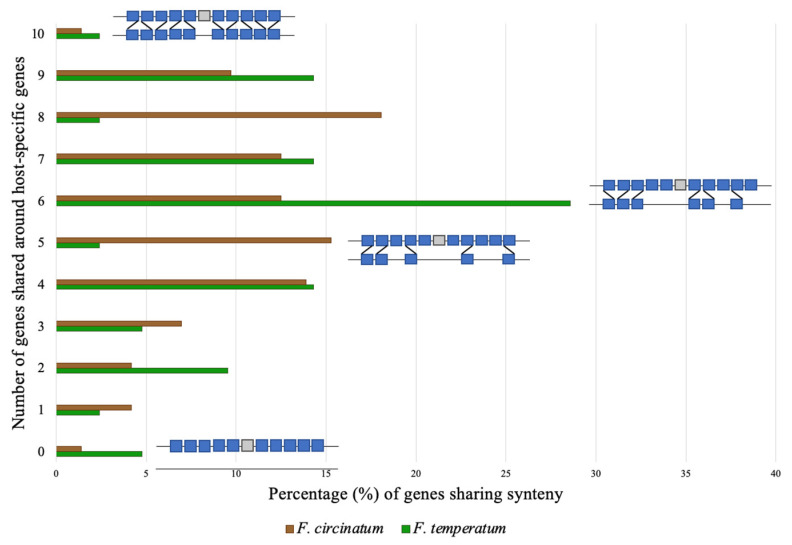
Data retrieved from the Shapiro–Wilk’s test for departure from normality for host-range-associated genes in *F. circinatum* and *F. temperatum*. Grey blocks represent the host-range-associated gene, whereas the blue blocks represent the genes downstream and upstream of the host-range-associated gene.

**Table 1 pathogens-11-00858-t001:** Genome statistics for the *Fusarium* genomes examined in this study.

*Fusarium* Species	Genome Size (bp)	Number of Scaffolds	Number of Chromosomes	Gene Density (ORFs/Mb ^1^)	Genes with Coding Sequences ^2^	GC Content (%)
*F. circinatum*	45,018,643	49	12	344.50	15,509	47.0
*F. fracticaudum*	46,252,763	50	12	352.28	16,294	47.6
*F. pininemorale*	47,778,485	153	12	332.35	15,879	46.0
*F. konzum*	43,487,959	2262	N/A	374.49	16,286	48.8
*F. subglutinans*	44,190,517	733	N/A	356.23	15,742	48.1
*F. temperatum*	45,458,781	43	12	341.67	15,532	47.0

^1^ ORFs/Mb = number of open reading frames per million base pairs. ^2^ As determined using MAKER2 [46].

**Table 2 pathogens-11-00858-t002:** Numbers of genes, and their inferred evolutionary origins, included in the pine- and grass-host-associated sets of host-range-associated genes.

Ancestral Origin Group ^1^	Number of Genes ^2^	
	*F. circinatum*	*F. temperatum*
1. FFSC	21 (10; 9)	13 (9)
2. FFSC and FOSC	6 (4; 4)	3 (1)
3. *Fusarium* but outside FFSC and FOSC	2 (0; 0)	2 (2)
4. Less than 10 NCBI BLAST hits and mostly *Fusarium*	10 (7; 5)	7 (5)
5. Less than 10 NCBI BLAST hits and mostly not *Fusarium*	2 (2; 2)	1 (1)
6. Outside *Fusarium*, but in the Ascomycetes	9 (7; 7)	5 (2)
7. Outside Fungi	1 (1; 1)	1 (1)
8. Unknown; no significant BLAST hit in the NCBI database	21 (7; 6)	15 (9)

^1^ Ancestral origin information from the phylogenetic trees in Appendix A. FFSC = *Fusarium fujikuroi* species complex and FOSC = *Fusarium oxysporum* species complex. ^2^ Number of host-range-associated genes with each type of phylogenetic origin. Pine-host-associated genes are from *F. circinatum*, and grass-host-associated genes are from *F. temperatum*. Values within parentheses indicate numbers of genes located in subtelomeric regions. For *F. circinatum* the value after the semicolons are numbers of genes for which RNAseq data are available from *F. circinatum* grown on media [51,52].

## Data Availability

Not applicable.

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
