# Peer review of "Characterization of Host-Specific Genes from Pine- and Grass-Associated Species of the Fusarium fujikuroi Species Complex"

_pathogens, 2022, doi:10.3390/pathogens11080858_

Round 1
Reviewer 1 Report
This manuscript used comparative genomics to make detailed analysis and characterization of host-specific genes from pine- and grass-as- sociated species of the Fusarium fujikuroi species complex. It provides sufficient support analysis to draw a conclusion that 72 and 47 genes identified as unique to the respective genome groups were potentially involved in diverse processes, ranging from transcription, regulation, and substrate transport, through to virulence/pathogenicity.
However, in the discussion section, these questions below should be addressed further.
What is the specific of host-range-associated genes involved in virulence?
To explain why host-range-associated genes tended to be located towards the subtelomeric regions of chromosomes?
What is the advantage of above allocation from evolution point of view?
Reviewer 2 Report
This manuscript was well organized and has potential interests to readers, which can be accepted for publication. But it still needs some revision before being formally accepted.
Fusarium fujikuroi species complexes are important plant pathogens and mycotoxin producing fungi, please add more description about Fusarium mycotoxin in introduction section, mainly involved in types of mycotoxins and their potential risk to food safety and human health.
I have labelled some parts in the file, where the sentences need re-write.
